# DP-SGD with weight clipping

## Abstract

Recently, due to the popularity of deep neural networks and other methods whose training typically relies on the optimization of an objective function, and due to concerns for data privacy, there is a lot of interest in differentially private gradient descent methods. To achieve differential privacy guarantees with a minimum amount of noise, it is important to be able to bound precisely the sensitivity of the information which the participants will observe. In this study, we present a novel approach that mitigates the bias arising from traditional gradient clipping. By leveraging a public upper bound of the Lipschitz value of the current model, we can achieve refined noise level adjustments. We present a new algorithm with improved differential privacy guarantees and a systematic empirical evaluation, showing that our new approach outperforms existing approaches also in practice.

## 1 Introduction

While machine learning allows for extracting statistical information from data with both high economical and societal value, there is a growing awareness of the risks for data privacy and confidentiality. Differential privacy (Dwork and Roth, 2013) has emerged as an important metric for studying statistical privacy.

Due to the popularity of deep neural networks (DNNs) and similar models, one of the recently most trending algorithmic techniques in machine learning has been stochastic gradient descent (SGD), which is a technique allowing for iteratively improving a candidate model using the gradient of the objective function on the data.

A popular class of algorithms to realize differential privacy while performing SGD is the DP-SGD algorithm (Abadi et al., 2016) and its variants. Essentially, these algorithms iteratively compute gradients, add differential privacy noise, and use the noisy gradient to update the model. To determine the level of differential privacy achieved, one uses an appropriate composition rule to bound the total information leaked in the several iterations.

To achieve differential privacy with a minimum amount of noise, it is important to be able to bound precisely the sensitivity of the information which the participants will observe. One approach is to bound the sensitivity of the gradient by assuming the objective function is Lipschitz continuous (Bassily et al., 2014). Various improvements exist in the case one can make additional assumptions about the objective function. For example, if the objective function is strongly convex, one can bound the number of iterations needed and in that way avoid to have to distribute the available privacy budget over too many iterations (Bassily et al., 2019). In the case of DNN, the objective function is not convex and typically not even Lipschitz continuous. Therefore, a common method is to 'clip' contributed gradients (Abadi et al., 2016), i.e., to divide gradients by the maximum possible norm they may get. These normalized gradients have bounded norm and hence bounded sensitivity.

In this paper, we argue that gradient clipping may not lead to optimal statistical results (see Section 4), and we propose instead to use weight clipping. Moreover, we also propose to consider the maximum gradient norm given the current position in the search space rather than the global maximum gradient norm, as this leads to additional advantages. In particular, our contributions are as follows:

- We introduce an novel approach, applicable to any feed-forward neural network, to compute gradient sensitivity that when applied in DP-SGD eliminates the need for gradient clipping. This strategy bridges the gap between Lipschitz-constrained neural networks and DP.

- We present a new algorithm, Lip-DP-SGD, that enforces bounded sensitivity of the gradients We argue that our approach, based on weight clipping, doesn't suffer from the bias which the classic gradient clipping can cause.

- We present an empirical evaluation, confirming that on a range of popular datasets our proposed method outperforms existing ones.

The remainder of this paper is organized as follows. First, we review a number of basic concepts, definitions and notations in Section 2. Next, we present our new method in Section 3 and present an empirical evaluation in Section 4. We discuss related work in Section 5. Finally, we provide conclusions and directions for future work in Section 6.

## 2 Preliminaries and background

In this section, we briefly review differential privacy, empirical risk minimization (ERM) and differentially private stochastic gradient descent (DP-SGD).

We will denote the space of all possible instances by $\mathcal{Z}$ and the space of all possible datasets by $\mathcal{Z}^*$. We will denote by $[N] = \{1 \dots N\}$ the set of the $N$ smallest positive integers.

### 2.1 Differential Privacy

An algorithm is differentially private if even an adversary who knows all but one instances of a dataset can't distinguish from the output of the algorithm the last instance in the dataset. More formally:

**Definition 2.1** (adjacent datasets). We say two datasets $Z_1, Z_2 \in \mathcal{Z}^*$ are adjacent, denoted $Z_1 \sim Z_2$, if they differ in at most one element. We denote by $\mathcal{Z}^*_\sim$ the space of all pairs of adjacent datasets.

**Definition 2.2** (differential privacy Dwork and Roth (2013)). Let $\epsilon > 0$ and $\delta > 0$. Let $\mathcal{A} : \mathcal{Z}^* \to \mathcal{O}$ be a randomized algorithm taking as input datasets from $\mathcal{Z}^*$. The algorithm $\mathcal{A}$ is $(\epsilon, \delta)$-differentially private $((\epsilon, \delta)$-DP) if for every pair of adjacent datasets $(Z_1, Z_2) \in \mathcal{Z}^*_\sim$, and for every subset $S \subseteq \mathcal{O}$ of possible outputs of $\mathcal{A}$, $P(\mathcal{A}(Z_1) \subseteq S) \leq e^\epsilon P(\mathcal{A}(Z_2) \subseteq S) + \delta$. If $\delta = 0$ we also say that $\mathcal{A}$ is $\epsilon$-DP.

If the output of an algorithm $\mathcal{A}$ is a real number or a vector, it can be privately released thanks to differential privacy mechanisms such as the Laplace mechanism or the Gaussian mechanism (Dwork et al., 2006). While our ideas are more generally applicable, in this paper we will focus on the Gaussian mechanism as it leads to simplier derivations. In particular, the Gaussian mechanism adds Gaussian noise to a number or vector which depends on its sensitivity on the input.

**Definition 2.3** (sensitivity). The $\ell_2$-sensitivity of a function $f : \mathcal{Z} \to \mathbb{R}^m$ is

$$s_2(f) = \max_{Z_1, Z_2 \in \mathcal{Z}^*_\sim} \|f(Z_1) - f(Z_2)\|_2$$

**Lemma 2.4** (Gaussian mechanism). *Let $f : \mathcal{Z} \to \mathbb{R}^m$ be a function. The Gaussian mechanism transforms $f$ into $\hat{f}$ with $\hat{f}(Z) = f(Z) + b$ where $b \sim \mathcal{N}(0, \sigma^2 I_m) \in \mathbb{R}^m$ is Gaussian distributed noise. If the variance satisfies $\sigma^2 \geq 2\ln(1.25/\delta)(s_2(f))^2/\epsilon^2$, then $\hat{f}$ is $(\epsilon, \delta)$-DP.*

### 2.2 Empirical risk minimization

Unless made explicit otherwise we will consider databases $Z = \{z_i\}_{i=1}^n$ containing $n$ instances $z_i = (x_i, y_i) \in \mathcal{X} \times \mathcal{Y}$ with $\mathcal{X} = \mathbb{R}^p$ and $\mathcal{Y} = \{0, 1\}$ sampled identically and independently (i.i.d.) from an unknown distribution on $\mathcal{Z}$. We are trying to build a model $f_\theta : \mathcal{X} \to \hat{\mathcal{Y}}$ (with $\hat{\mathcal{Y}} \subseteq \mathbb{R}$) parameterized by $\theta \in \Theta \subseteq \mathbb{R}^m$,

so it minimizes the expected loss $\mathcal{L}(\theta) = \mathbb{E}_z[\mathcal{L}(\theta; z)]$, where $\mathcal{L}(\theta; z) = \ell(f_\theta(x), y)$ is the loss of the model $f_\theta$ on data point z. One can approximate $\mathcal{L}(\theta)$ by

$$\hat{R}(\theta; Z) = \frac{1}{n}\sum_{i=1}^{n}\mathcal{L}(\theta; z_i) = \frac{1}{n}\sum_{i=1}^{n}\ell(f_\theta(x_i), y_i),$$

the empirical risk of model $f_\theta$. Empirical Risk Minimization (ERM) then minimizes an objective function $F(\theta, Z)$ which adds to this empirical risk a regularization term $\psi(\theta)$ to find an estimate $\hat{\theta}$ of the model parameters:

$$\hat{\theta} \in \underset{\theta \in \Theta}{\arg\min}\; F(\theta; Z) := \hat{R}(\theta; Z) + \gamma\psi(\theta)$$

where $\gamma \geq 0$ is a trade-off hyperparameter.

**Feed forward neural networks** An important and easy to analyze class of neural networks are the feed forward networks (FNN). A FNN is a direct acyclic graph where connections between nodes don't form cycles.

**Definition 2.5.** A FNN $f_\theta : \mathbb{R}^n \to \mathbb{R}^m$ is a function which can be expressed as

$$f_\theta = f_{\theta_K}^{(K)} \circ \ldots \circ f_{\theta_1}^{(1)}$$

where $f_{\theta_k}^{(k)} : \mathbb{R}^{n_k} \to \mathbb{R}^{n_{k+1}}$. $f_{\theta_k}^{(k)}$ is the $k$-th layer function parameterized by $\theta_k$ for $1 \leq k \leq K$. We denote the input of $f_{\theta_k}^{(k)}$ by $x_k$ and its output by $x_{k+1}$. Here, $\theta = (\theta_1 \ldots \theta_K)$, $n = n_1$ and $m = n_{K+1}$.

Common layers include fully connected layers, convolutional layers and activation layers. Parameters of the first two correspond to weight and bias matrices, $\theta_k = (W_k, B_k)$, while activation layers have no parameter, $\theta_k = ()$.

## 2.3 Stochastic gradient descent

To minimize $F(\theta, Z)$, one can use gradient descent, i.e., iteratively for a number of time steps $t = 1 \ldots T$ one computes a gradient $g^{(t)} = \nabla F(\tilde{\theta}^{(t)}, Z)$ on the current model $\tilde{\theta}^{(t)}$ and updates the model setting $\tilde{\theta}^{(t+1)} = \tilde{\theta}^{(t)} - \eta(t)g^{(t)}$ where $\eta(t)$ is a learning rate. Stochastic gradient descent (SGD) introduces some randomness and avoids the need to recompute all gradients in each iteration by sampling in each iteration a batch $V \subseteq Z$ and computing an approximate gradient $\hat{g}_t = \frac{1}{|V|}\left(\sum_{i=1}^{|V|}\nabla\mathcal{L}(\tilde{\theta}^{(t)}, v_i) + b^{(t)}\right) + \gamma\nabla\psi(\theta)$.

To avoid leaking sensitive information, Abadi et al. (2016) proposes to add noise to the gradients. Determining good values for the scale of this noise has been the topic of several studies. One simple strategy starts by assuming an upper bound for the norm of the gradient. Let us first define Lipschitz functions:

**Definition 2.6** (Lipschitz function). Let $L^g > 0$. A function $f$ is $L^g$-Lipschitz with respect to some norm $\|\cdot\|$ if for all $\theta, \theta' \in \Theta$ there holds $\|f(\theta) - f(\theta')\| \leq L^g \|\theta - \theta'\|$. If $f$ is differentiable, by Rademacher's theorem the above property is equivalent to:

$$\|\nabla f(\theta)\| \leq L^g, \quad \forall\theta \in \Theta$$

We call the smallest value $L^g$ for which $f$ is $L^g$-Lipschitz the Lipschitz value of $f$.

Then, from the model one can derive a constant $L^g$ such that the objective function is $L^g$-Lipschitz, while knowing bounds on the data next allows for computing a bound on the sensitivity of the gradient. Once one knows the sensitivity, one can determine the noise to be added from the privacy parameters as in Lemma 2.4. The classic DP-SGD algorithm (Abadi et al., 2016), which we recall in Algorithm 2 in Appendix A for completeness, clips the gradient of each instance to a maximum value $C$ (i.e., scales down the gradient if its norm is above $C$) and then adds noise based on this maximal norm $C$.

$$\tilde{g}_t = \frac{1}{|V|}\left(\sum_{i=1}^{|V|}\mathrm{clip}_C\left(\nabla_{\tilde{\theta}}\mathcal{L}\left(\tilde{\theta}^{(t)}, v_i\right)\right) + b_t\right) + \gamma\nabla\psi(\theta)$$

where $b_t$ is appropriate noise and where

$$\text{clip}_C(v) = v . \min\left(1, \frac{C}{\|v\|}\right).$$

## 2.4 Regularization

Several papers (Ioffe and Szegedy, 2015; Wu and He, 2018) have pointed out that regularization can help to improve the performance of stochastic gradient descent. Although batch normalization (Ioffe and Szegedy, 2015) does not provide protection against privacy leakage, group normalization (Wu and He, 2018) has the potential to do so (De et al., 2022). De et al. (2022) combines group normalization with DP-SGD, the algorithm to which we propose an improvement in the current paper. Group normalization is a technique adding specific layers, called group normalization layers, to the network. Making abstraction of some elements specific to image datasets, we can formalize it as follows.

For a vector $v$, we will denote the dimension of $v$ by $|v|$, i.e., $v \in \mathbb{R}^v$.

If the $k$-th layer is a normalization layer, then there holds $|x_k| = |x_{k+1}|$. Moreover, the structure of the normalization layer defines a partitioning $\Gamma_k = \{\Gamma_{k,1} \ldots \Gamma_{k,|G|}\}$ of $[|x_k|]$, i.e., a partitioning of the components of $x_k$. The components of $x_k$ and $x_{k+1}$ are then grouped, and we define $x_k^{(k:q)} = (x_{k,j})_{j \in \Gamma_{k,q}}$, i.e., $x_k^{(k:q)}$ is a subvector containing a group of components. Similarly, $x_{k+1}^{(k:q)} = (x_{k+1,j})_{j \in \Gamma_{k,q}}$. Then, the $k$-th layer performs the following operation:

$$x_{k+1}^{(k:q)} = f_{\theta_k}^{(k)}(x_k^{(k:q)}) = \frac{1}{\sigma^{(k:q)}}\left(x_k^{(k:q)} - \mu^{(k:q)}\right), \tag{1}$$

(but note we will adapt this in Eq (2)) where

$$\mu^{(k:q)} = \frac{1}{|\Gamma_{k,q}|}\sum_{j=1}^{|\Gamma_{k,q}|} x_{k,j},$$

$$\sigma^{(k:q)} = \left(\frac{1}{|\Gamma_{k,q}|}\sum_{j=1}^{|\Gamma_{k,q}|}\left(x_{k,j} - \mu^{(k:q)}\right)^2 + \kappa\right)^{1/2},$$

with $\kappa$ a small constant.

Various feature normalization methods primarily vary in their definitions of the partition of features $\Gamma_{k,q}$.

## 3 Our approach

In this work, we constrain the objective function to be Lipschitz, and exploit this to determine sensitivity. An important advantage is that while traditional DP-SGD controls sensitivity via gradient clipping of each sample separately, our new method estimates gradient sensitivity based on only the model in a data-independent way. This is grounded in Lipschitz-constrained model literature (Section 5), highlighting the connection between the Lipschitz value for input and parameter. Subsection 3.1 delves into determining an upper Lipschitz bound. Subsection 3.2 demonstrates the use of backpropagation for gradient sensitivity estimation, and in 3.3, we introduce Lip-DP-SGD, a novel algorithm ensuring privacy without gradient clipping.

### 3.1 Estimating lipschitz values

In this section we bound Lipschitz values of different types of layers. We treat linear operations (e.g., linear transformations, convolutions) and activation functions as different layers. We present results with regard to any $p$-norms with $p \in \{1, 2, +\infty\}$.

**Loss function and activation layer.** Examples of Lipschitz losses encompass Softmax Cross-entropy, Cosine Similarity, and Multiclass Hinge. When it comes to activation layers, layers composed of an activation

function, several prevalent ones, such as ReLU, tanh, and Sigmoid are 1-Lipschitz with respect to all $p$-norms. We provide a detailed list in the Appendix Table 2.

**Normalization layer.** To be able to easily bound sensitivity, we define the operation of a normalization layer $f_{\theta_k}^{(k)}$ slightly differently than Eq (1):

$$x_{k+1}^{(k:q)} = f_{\theta_k}^{(k)}(x_k^{(k:q)}) = \frac{x_k^{(k:q)} - \mu^{(k:q)}}{\max(1, \sigma^{(k:q)})}. \tag{2}$$

It is easy to see that the sensitivity is bounded by

$$\left\| \frac{\partial f_{\theta_k}^{(k)}}{\partial x_k} \right\|_p \leq \max_{q \in [|\Gamma_k|]} \frac{1}{\max\left(1, \sigma^{(k:q)}\right)} \leq 1. \tag{3}$$

Note that a group normalization layer has no trainable parameters.

**Linear layers.** Gouk et al. (2020) presents a formulation for the Lipschitz value of linear layers in relation to p-norms. This formulation can be extended to determine the Lipschitz value with respect to the parameters. Therefore, if $f_{\theta_k}^{(k)}$ is a linear layer, then

$$\left\| \frac{\partial f_{\theta_k}^{(k)}}{\partial \theta_k} \right\|_p = \left\| \frac{\partial(W_k^\top x_k + B_k)}{\partial(W_k, B_k)} \right\|_p = \|(\vec{x_k}, 1)\|_p,$$
$$\left\| \frac{\partial f_{\theta_k}^{(k)}}{\partial x_k} \right\|_p = \left\| \frac{\partial(W_k^\top x_k + B_k)}{\partial x_k} \right\|_p = \|W_k\|_p, \tag{4}$$

with $\vec{x_k}$ the serialized vector of $x_k$.

**Convolutional layers.** There are many types of convolutional layers, e.g., depending on the data type (strings, 2D images, 3D images ...), shape of the filter (rectangles, diamonds ...). Here we provide as an example only a derivation for convolutional layers for 2D images with rectangular filter. In that case, the input layer consists of $n_k = c_{in}hw$ nodes and the output layer consists of $n_{k+1} = c_{out}hw$ nodes with $c_{in}$ input channels, $c_{out}$ output channels, $h$ the height of the image and $w$ the width. Then, $\theta_k \in \mathbb{R}^{c_{in} \times c_{out} \times h' \times w'}$ with $h'$ the height of the filter and $w'$ the width of the filter. Indexing input and output with channel and coordinates, i.e., $x_k \in \mathbb{R}^{c_{in} \times h \times w}$ and $x_{k+1} \in \mathbb{R}^{c_{out} \times h \times w}$ we can then write

$$x_{k+1,c,i,j} = \sum_{d=1}^{c_{in}} \sum_{r=1}^{h'} \sum_{s=1}^{w'} x_{k,d,i+r,j+s} \theta_{k,c,d,r,s}$$

where components out of range are zero. We can derive (see Appendix B.1 for details) that

$$\left\| \frac{\partial f_{\theta_k}^{(k)}}{\partial x_k} \right\|_p \leq (h'w')^{\frac{1}{p}} \|\theta_k\|_p \tag{5}$$

$$\left\| \frac{\partial f_{\theta_k}^{(k)}}{\partial \theta_k} \right\|_p \leq (h'w')^{\frac{1}{p}} \|\vec{x_k}\|_p \tag{6}$$

**Residual connections** Resnet architectures (He et al., 2016) are usually based on residual blocks:

$$f_{\theta_{j+k}}^{(j+k)}(x_j) = x_j + (f_{\theta_{j+k-1}}^{(j+k-1)} \circ \ldots \circ f_{\theta_j}^{(j)})(x_j)$$

Gouk et al. (2020) shows that the Lipschitz value of residual blocks is bounded by

$$\left\| \frac{\partial f_{\theta_{k+j}}^{(k+j)}}{\partial x_j} \right\|_p \leq 1 + \prod_{i=j}^{j+k-1} \left\| \frac{\partial f_{\theta_i}^{(i)}}{\partial x_i} \right\|_p. \tag{7}$$

We summarize the upper bounds of the Lipschitz values, either on the input or on the parameters, for each layer type in the Appendix Table 2. We can conclude that networks for which the norms of the parameter vectors $\theta_k$ are bounded, are Lipschitz networks as introduced in Miyato et al. (2018), i.e., they are FNN for which each layer function $f_{\theta_k}^{(k)}$ is Lipschitz. We will denote by $\Theta_{\leq C}$ and by $\Theta_{=C}$ the sets of all paremeter vectors $\theta$ for $f_\theta$ such that $\|\theta_k\| \leq C$ and $\|\theta_k\| = C$ respectively, for $k = 1 \ldots K$.

## 3.2 Backpropagation

Consider a feed-forward network $f_\theta$. We define $\mathcal{L}_k(\theta, (x_k, y)) = \ell\left(\left(f_{\theta_K}^{(K)} \circ \ldots \circ f_{\theta_k}^{(k)}\right)(x_k), y\right)$. For feed-forward networks, the chain rule gives:

$$\frac{\partial \mathcal{L}_k}{\partial x_k} = \frac{\partial \ell}{\partial x_{K+1}} \frac{\partial f_{\theta_K}^{(K)}}{\partial x_k}. \tag{8}$$

Any matrix or vector norm is submultiplicative, especially the $\|\cdot\|_p$ norm with $p \in \{1, 2, +\infty\}$, hence:

$$\left\| \frac{\partial \mathcal{L}_k}{\partial x_k} \right\|_p \leq \left\| \frac{\partial \ell}{\partial x_{K+1}} \right\|_p \left\| \frac{\partial f_{\theta_K}^{(K)}}{\partial x_k} \right\|_p. \tag{9}$$

In Section 3.1, we show that the Lipschitz value with regard to the input of $f_{\theta_k}^{(k)}$ is bounded and the bound depends on the norm of the parameters in the case of linear or convolutional layers. Let $c_k$ be the Lipschitz value with regards to the input of any layer $k$. As $f_{\theta_k}$ is Lipschitz constrained, when the layer $k$ has parameters, $c_k$ is a linear function of $\min(C, \|\theta_k\|)$, as stated in Equations (4) and (5), with $C$ the maximum weight norm. If $\left\| \frac{\partial \ell}{\partial x_{K+1}} \right\|_p \leq \tau$, where $\ell$ represents the loss then,

$$\left\| \frac{\partial \mathcal{L}_k}{\partial x_k} \right\|_p \leq \tau \prod_{i=k}^{K} c_i. \tag{10}$$

We now consider the induced $\|\cdot\|_{2,p}$ norm ($\|A\|_{\alpha,\beta} = \sup_{x \neq 0} \frac{\|Ax\|_\beta}{\|x\|_\alpha}$) for studiying $\frac{\partial \mathcal{L}_k}{\partial \theta_k}$. Induced norms are consistent, then one can prove that $\|AB\|_{\alpha,\gamma} \leq \|A\|_{\beta,\gamma} \|B\|_{\alpha,\beta}$ (Cape et al., 2017) which, applied to the chain rule, gives:

$$\left\| \frac{\partial \mathcal{L}_k}{\partial \theta_k} \right\|_{2,p} \leq \left\| \frac{\partial \mathcal{L}_{k+1}}{\partial x_{k+1}} \right\|_p \left\| \frac{\partial f_{\theta_k}^{(k)}}{\partial \theta_k} \right\|_{2,p} \tag{11}$$

Combining 10 and 11 provides an upper bound of the $2, p$-norm of the gradient at layer $k$,

$$\left\| \frac{\partial \mathcal{L}_k}{\partial \theta_k} \right\|_{2,p} \leq \tau \prod_{i=k+1}^{K} c_i \left\| \frac{\partial f_{\theta_k}^{(k)}}{\partial \theta_k} \right\|_{2,p} \tag{12}$$

If $\theta_k \neq \emptyset$, then Equations (4) and (6) show that the upper bound of $\left\| \frac{\partial f_{\theta_k}^{(k)}}{\partial \theta_k} \right\|_{2,p}$ depends on $\|\vec{x_k}\|_2$. Hence,

$$\left\| \frac{\partial \mathcal{L}_k}{\partial \theta_k} \right\|_{2,p} \leq \tau \prod_{i=k+1}^{K} c_i X_k, \tag{13}$$

with $X_k = \sqrt{hw} \max_{x_k \in V_k} \|\vec{x_k}\|_2$ with $hw = 1$ in case of a linear layer and $V_k = \{(f_k \circ \ldots \circ f_1)(x)|(x,y) \in V\}$.

### 3.3 Lip-DP-SGD

We introduce in Algorithm 1 a novel differentially private stochastic gradient descent algorithm, called Lip-DP-SGD, that leverages the estimation of the per-layer sensitivity of the model to provide differential privacy without gradient clipping.

---

**Algorithm 1** LIP-DP-SGD: Differentially Private Stochastic Gradient Descent with Lipschitz constrains.

---

1: **Input:** Data set $Z \in \mathcal{Z}^*$, feed-forward model $f_\theta$, loss function $\ell$ with Lipschitz value $\tau$, hypothesis space $\Theta \subseteq \mathbb{R}^k$, number of epochs $T$, noise multiplier $\sigma$, batch size $s \geq 1$, learning rate $\eta$, norm $p \in \{1, 2, \infty\}$, max weight norm $C$.
2: Initialize $\tilde{\theta}$ randomly from $\Theta$
3: $(c_k, \tilde{\theta}_k)_{k=1}^K \leftarrow \text{CLIPWEIGHTS}(\tilde{\theta}, C, p)$
4: **for** $t \in [T]$ **do**
5:     $(\Delta_k)_{k=1}^K \leftarrow (\tau \prod_{i=k}^K c_i)_{k=1}^K$                                    ▷ Lipschitz value at layer $k$, see Equation (5)
6:     $V \leftarrow \emptyset$                                                       ▷ Poisson sampling
7:     **while** $V = \emptyset$ **do**
8:         **for** $z \in Z$ **do**
9:             With probability $s/|Z|$: $V \leftarrow V \cup \{z\}$
10:         **end for**
11:     **end while**
12:     **for** $k = 1 \ldots K$ **do**                                            ▷ Compute gradient per layer
13:         $X_k \leftarrow \max\limits_{i \in |V|} \|f_{\tilde{\theta}_{k-1}}(x_i)\|_2$                       ▷ Max input norm of layer $k$, see Equation (13)
14:         Draw $b_k \sim \mathcal{N}(0, \sigma^2 \Delta_k^2 \mathbb{I})$
15:         $\tilde{g}_k \leftarrow \frac{1}{|V|}\left(\frac{1}{X_k}\sum_{i=1}^{|V|}\nabla_{\tilde{\theta}_k}\ell(f_{\tilde{\theta}}(x_i), y_i) + b_k\right)$              ▷ DP gradient, see Equation (14)
16:         $\tilde{\theta}_k \leftarrow \tilde{\theta}_k - \eta(t)\tilde{g}_k$                                              ▷ Update
17:     **end for**
18:     $(c_k, \tilde{\theta}_k)_{k=1}^K \leftarrow \text{CLIPWEIGHTS}(\tilde{\theta}, C, p)$                           ▷ Enforce Lipschitzness
19: **end for**
20: **Output:** $\tilde{\theta}$ and compute $(\epsilon, \delta)$ with privacy accountant.
21: **function** CLIPWEIGHTS($\tilde{\theta}, C$)
22:     **for** $k = 1 \ldots K$ **do**
23:         **if** $\theta_k \neq \emptyset$ **then**
24:             $c_k \leftarrow \min(C, \|\tilde{\theta}_k\|_p)$
25:             $\tilde{\theta}_k \leftarrow c_k \tilde{\theta}_k / \|\tilde{\theta}_k\|_p$
26:         **else**
27:             $c_k \leftarrow 1$
28:         **end if**
29:     **end for**
30:     **return** $(c_k, \tilde{\theta}_k)_{k=1}^K$
31: **end function**

---

**Differential Privacy.** From Equation (13) follows,

$$\forall (x, y) \in \mathcal{X} \times \mathcal{Y}, \quad \left\|\frac{\nabla_{\tilde{\theta}_k}\ell(f_{\tilde{\theta}}(x), y)}{X_k}\right\|_{2,p} \leq \tau \prod_{i=k+1}^K c_i \tag{14}$$

The left-hand side of Equation (14) represents the gradient scaled by the maximum input norm. The $\ell_{2,p}$-sensitivity of this scaled gradient is bounded, with the upper-bound depending solely on the Lipschitz constant of the loss function $\tau$, the maximum parameter norm $C$, and/or the parameter norm $\|\tilde{\theta}_k\|_{2,p}$.

**Theorem 3.1.** *Given a feed-forward model $f_\theta$ composed of Lipschitz constrained operators and a Lipschitz loss $\ell$, LIP-DP-SGD is differentially private.*

Indeed, the scaled gradient's sensitivity is determined without any privacy costs, as it depends only on the current parameter values (which are privatized in the previous step, and post-processing privatized values doesn't take additional privacy budget) and not on the data. If the selected norm is the $2, 2$-norm, the Gaussian mechanism can be applied to ensure privacy by utilizing the sensitivity of the scaled gradient. Otherwise, the exponential mechanism can be employed to achieve differential privacy (McSherry and Talwar, 2007) using a custom score function.

**Privacy accounting.** Lip-DP-SGD adopts the same privacy accounting as DP-SGD. Specifically, the accountant draws upon the privacy amplification Kasiviswanathan et al. (2011) brought about by Poisson sampling and the Gaussian moment accountant Abadi et al. (2016). It's worth noting that while we utilized the Renyi Differential Privacy (RDP) accountant Abadi et al. (2016); Mironov et al. (2019) in our experiments, Lip-DP-SGD is versatile enough to be compatible with alternative accountants. Note that RDP is primarily designed to operate with the Gaussian mechanism, but Wang et al. (2019) demonstrate its applicability with other differential privacy mechanisms, particularly the exponential mechanism.

**Requirements.** As detailed in Section 3.1, the loss and the model operators need to be Lipschitz. We've enumerated several losses and operators that meet this criterion in the Appendix Table 2. While we use CLIPWEIGHTS to characterize Lipschitzness Yoshida and Miyato (2017); Miyato et al. (2018) in our study 3.1, other methods are also available, as discussed in Arjovsky et al. (2017).

**ClipWeights.** The CLIPWEIGHTS function is essential to the algorithm, ensuring Lipschitzness, which facilitates model sensitivity estimation. As opposed to standard Lipschitz-constrained networks Yoshida and Miyato (2017); Miyato et al. (2018) which increase or decrease the norms of parameters to make them equal to a pre-defined value, our approach normalizes weights only when their current norm exceeds a threshold. This results in adding less DP noise for smaller norms. Importantly, as $\tilde{\theta}$ is already made private in the previous iteration, its norm is private too. Note that, when the norm used is the 2-norm i.e., CLIPWEIGHTS is a spectral normalization, we perform also a Björck orthogonalization for for fast and near-orthogonal convolutions (Li et al., 2019).

**Computing norms** The $\ell_{2,1}$ and $\ell_{2,\infty}$ norms can be computed exactly in linear time relative to the number of elements. However, calculating the $\ell_{2,2}$ norm, which corresponds to the largest singular value of the matrix, is infeasible using standard singular value decomposition techniques. To address this efficiently, we use techniques based on power method (Gouk et al., 2020; Scaman and Virmaux, 2019) that leverage the backward computational graph of modern deep learning libraries like (Abadi et al., 2015, TensorFlow) and (Paszke et al., 2019, PyTorch).

### 3.4 Avoiding the bias of gradient clipping

Our Lip-DP-SGD algorithm finds a local optimum (for $\theta$) of $F(\theta, Z)$ in $\Theta_{\leq C}$ while DP-SGD doesn't necessarily find a local optimum of $F(\theta, Z)$ in $\Theta$. In particular, we prove in Appendix E the following

**Theorem 3.2.** *Let $F$ be an objective function as defined in Section 2.2, and $Z$, $f_\theta$, $\mathcal{L}$, $\Theta = \Theta_{\leq C}$, $T$, $\sigma = 0$, $s$, $\eta$ and $C$ be input parameters of Lip-DP-SGD satisfying the requirements specified in Section 3.3. Assume that for these inputs Lip-DP-SGD converges to a point $\theta^*$ (in the sense that $\lim_{k,T \to \infty} \theta_k = \theta^*$). Then, $\theta^*$ is a local optimum of $F(\theta, Z)$ in $\Theta_{\leq C}$.*

Essentially, making abstraction of the unbiased DP noise, the effect of scaling weight vectors to have bounded norm after a gradient step is equivalent to projecting the gradient on the boundary of the feasible space if the gradient brings the parameter vector out of $\Theta_{\leq C}$.

Furthermore, Chen et al. (2020) shows an example showing that gradient clipping can introduce bias. We add a more detailed discussion in Appendix E. Hence, DP-SGD does not necessarily converge to a local optimum of $F(\theta, Z)$, even when sufficient data is available to estimate $\theta$. While Lip-DP-SGD can only find models in $\Theta_{\leq C}$ and this may introduce another suboptimality, as our experiments will show this is only a minor drawback in practice, while also others observed that Lipschitz networks have good properties Béthune et al.

(2023). Moreover, it is easy to check whether Lip-DP-SGD outputs parameters on the boundary of $\Theta_{\leq C}$ and hence the model could potentially improve by relaxing the weight norm constraint. In contrast, it may not be feasible to detect that DP-SGD is outputting potentially suboptimal parameters. Indeed, consider a federated learning setting (e.g., Bonawitz et al. (2017)) where data owners collaborate to compute a model without revealing their data. Each data owner locally computes a gradient and clips it, and then the data owners securely aggregate their gradients and send the average gradient to a central party updating the model. In such setting, for privacy reasons no party would be able to evaluate that gradient clipping introduces a strong bias in some direction. Still, our experiments show that in practice at the time of convergence for the best hyperparameter values clipping is still active for a significant fraction of gradients (See Appendix C.5)

## 4 Experimental results

In this section, we conduct an empirical evaluation of our approach.

### 4.1 Experimental setup

We consider the following experimental questions:

Q1 How does Lip-DP-SGD, our proposed technique, compare against the conventional DP-SGD as introduced by Abadi et al. (2016)?

Q2 What is the effect of allowing $\|\theta_k\| < C$ rather than normalizing $\|\theta_k\|$ to $C$? This question seems relevant given that some authors (e.g., Béthune et al. (2023)) also suggest to consider networks which constant gradient norm rather than maximal gradient norm, i.e., roughly with $\theta$ in $\Theta_{=C}$ rather than $\Theta_{\leq C}$.

**Norms.**  In this section, we focus on the $\ell_{2,2}$ norm. All subsequent results are based on this norm. Although computing this norm is more computationally intensive, it allows for the use of the Gaussian mechanism, which is standard in this field.

**Hyperparameters.**  We selected a number of hyperparameters to tune for our experiments, aiming at making a fair comparison between the studied techniques while minimizing the distractions of potential orthogonal improvements. To optimize these hyperparameters, we used Bayesian optimization Balandat et al. (2020). Appendix C.1 provides a detailed discussion.

**Datasets and models.**  We carried out experiments on both tabular datasets and datasets with image data. First, we consider a collection of 7 real-world tabular datasets (names and citations in Table 1). For these, we trained multi-layer perceptrons (MLP). A comprehensive list of model-dataset combinations is available in the Appendix Table 4. To answer question Q2, we also implemented Fix-Lip-DP-SGD, a version of Lip-DP-SGD limited to networks whose weight norms are fixed, i.e., $\forall k : \|\theta_k\| = C$, obtained by setting $u_k^{(\theta)} \leftarrow C$ in Line 24 in Algorithm 1.

Second, the image datasets used include MNIST (Deng, 2012), Fashion-MNIST (Xiao et al., 2017), and CIFAR-10 (Krizhevsky et al., 2009). We trained convolutional neural networks (CNNs) for the first two datasets and a Wide-ResNet Zagoruyko and Komodakis (2016) for the latter (see details in Table 4). We implemented all regularization techniques as described in De et al. (2022), including group normalization (Wu and He, 2018), large batch size, weight standardization (Qiao et al., 2020), augmentation multiplicity (De et al., 2022), and parameter averaging (Polyak and Juditsky, 1992). These techniques are compatible with Lipschitz-constrained networks, except for group normalization, for which we proposed an adapted version in Equation (2).

We opted for the accuracy to facilitate easy comparisons with prior research.

**Infrastructure.**  All experiments were orchestrated across dual Tesla P100 GPU platforms (12GB capacity), operating under CUDA version 10, with a 62GB RAM provision for Fashion-MNIST and CIFAR-10. Remaining experiments were performed on an E5-2696V2 Processor setup, equipped with 8 vCPUs and a 52GB RAM

cache. The total runtime of the experiments was approximately 50 hours, which corresponds to an estimated carbon emission of 1.96 kg Lacoste et al. (2019). More details on the experimental setup and an analysis of the runtime can be found in Appendix C.

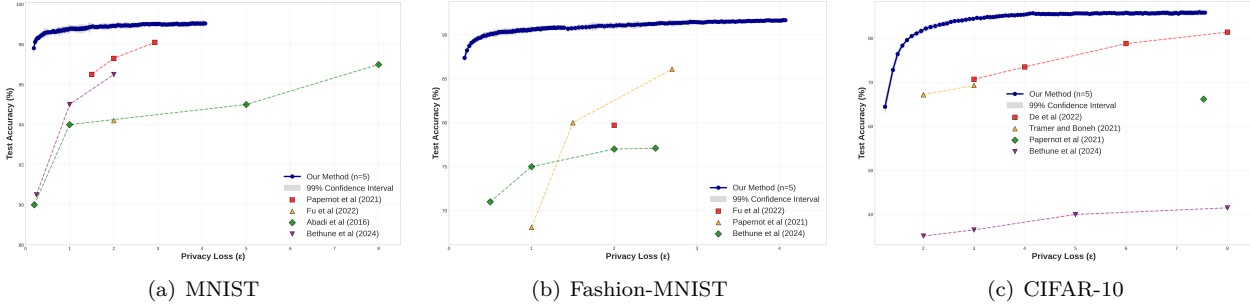

(a) MNIST        (b) Fashion-MNIST        (c) CIFAR-10

Figure 1: Accuracy results, with a fixed $\delta = 10^{-5}$, for the MNIST (1(a)), Fashion-MNIST (1(b)), and CIFAR-10 (1(c)) test datasets. The plots show the median accuracy over 5 runs, with vertical lines indicating the standard error of the mean. See Appendix C.1 for details on model specifications and hyperparameters.

Table 1: Accuracy and $\epsilon$ per dataset and method at $\delta = 1/n$, in bold the best result and underlined when the difference with the best result is not statistically significant at a level of confidence of 5%.

| Methods | | DP-SGD | Lip-DP-SGD | Fix-Lip-DP-SGD |
|---|---|---|---|---|
| Datasets (#instances $n$ × #features $p$) | $\epsilon$ | | | |
| Adult Income (48842x14) Becker et al. (1996) | 0.414 | 0.824 | **0.831** | 0.804 |
| Android (29333x87) Mathur et al. (2022) | 1.273 | 0.951 | **0.959** | 0.945 |
| Breast Cancer (569x32) Wolberg et al. (1995) | 1.672 | 0.773 | **0.798** | 0.7 |
| Default Credit (30000x24) Yeh (2016) | 1.442 | 0.809 | **0.816** | 0.792 |
| Dropout (4424x36) Realinho et al. (2021) | 1.326 | 0.763 | **0.819** | 0.736 |
| German Credit (1000x20) Hofmann (1994) | 3.852 | 0.735 | 0.746 | 0.768 |
| Nursery (12960x8) Rajkovic (1997) | 1.432 | 0.919 | **0.931** | 0.89 |

## 4.2 Results

**Image datasets.** In Figure 1, Lip-DP-SGD outperforms all state-of-the-art results on all three image datasets. Previous performances were based on DP-SGD as introduced by Abadi et al. (2016), either combined with regularization techniques (De et al., 2022) or with bespoke activation functions (Papernot et al., 2021). Note that while we present results using the same CNN model for MNIST and Fashion-MNIST, the results for CIFAR-10 come from a Wide-ResNet-18-4. See the complete list of hyperparameters and settings in Appendix C.1.

In addition to state-of-the-art baselines, we report results from Bethune et al. (2023), whose approach also relies on Lipschitz-constrained networks to ensure differential privacy. Unlike their method, our approach bypasses the dependance on worst-case input norms at each layer. As a result, we achieve up to twice the performance on challenging tasks such as CIFAR-10.

**Tabular datasets.** In Table 1, we perform a Wilcoxon Signed-rank test, at a confidence level of 5%, on 10 measures of accuracy for each dataset between the DP-SGD based on the gradient clipping and the Lip-DP-SGD based on our method. Lip-DP-SGD consistently outperforms DP-SGD in terms of accuracy. This trend holds across datasets with varying numbers of instances and features, including tasks with imbalanced datasets like Dropout or Default Credit datasets. While highly impactful for convolutional layers, group normalization does not yield improvements for either DP-SGD or Lip-DP-SGD in the case of tabular datasets. See Appendix C.4 for complete results.

Additionally, Table 1 presents the performance achieved by constraining networks to Lipschitz networks, where the norm of weights is set to a constant, denoted as Fix-Lip-DP-SGD. The results from this approach are inferior, even when compared to DP-SGD.

**Conclusion.** In summary, our experimental results demonstrate that Lip-DP-SGD sets new state-of-the-art benchmarks on the three most popular vision datasets, outperforming DP-SGD. Additionally, Lip-DP-SGD also outperforms DP-SGD on tabular datasets using MLPs, where it is advantageous to allow the norm of the weight vector $\theta$ to vary rather than normalizing it to a fixed value, leveraging situations where it can be smaller.

## 5  Related Work

**DP-SGD.** DP-SGD algorithms have been developped to guarantee privacy on the final output Chaudhuri et al. (2011), on the loss function Kifer et al. (2012) or on the publishing of each gradient used in the descent Bassily et al. (2014); Abadi et al. (2016).

To keep track of the privacy budget consumption, Bassily et al. (2014) relies on the strong composition theorem Dwork et al. (2010) while Abadi et al. (2016) is based on the moment accountant and gives much tighter bounds on the privacy loss than Bassily et al. (2014).

This has opened an active field of research that builds upon Abadi et al. (2016) in order to provide better estimation of the hyperparameters e.g., the clipping norm McMahan et al. (2017); Andrew et al. (2022), the learning rate Koskela and Honkela (2020), or the step size of the privacy budget consumption Lee and Kifer (2018); Chen and Lee (2020); Yu et al. (2019); or to enhance performance with regularization techniques De et al. (2022). Gradient clipping remains the standard approach, and most of these ideas can be combined with our improvements.

**Lipschitz continuity.** Lipschitz continuity is an essential requirement for differential privacy in some private SGD algorithms Bassily et al. (2014). However, since deep neural networks (DNNs) have an unbounded Lipschitz value Scaman and Virmaux (2019), it is not possible to use it to scale the added noise. Several techniques have been proposed to enforce Lipschitz continuity to DNNs, especially in the context of generative adversarial networks (GANs) Miyato et al. (2018); Gouk et al. (2020). These techniques, which mainly rely on weight clipping, can be applied to build DP-SGD instead of the gradient clipping method, as described in Section 3 discusses how Bethune et al. (2023) proposes several concepts related to our Fix-Lip-DP-SGD variant. However, their use of an upper bound on the input norm limits their ability to achieve competitive results and extend beyond the 2-norm. In contrast, our paper demonstrates that Lip-DP-SGD outperforms Fix-Lip-DP-SGD (Table 1), showcases the strong synergy between weight normalization and regularization techniques (see Figure 1), enables the theoretical use of the $\ell_{2,1}$ and the $\ell_{2,\infty}$-norms, and illustrates how the scaled gradient (Equation (14)) we employ unlocks high-accuracy for differentially private machine learning.

## 6  Conclusion and discussion

In this paper we proposed a new differentially private stochastic gradient descent algorithm without gradient clipping. We derived a methodology to estimate the gradient sensitivity to scale the noise. An important advantage of weight clipping over gradient clipping is that it avoids the bias introduced by gradient clipping and the algorithm converges to a local optimum of the objective function. We showed empirically that this yields a significant improvement in practice and we argued that this approach circumvent the bias induced by classical gradient clipping.

Several opportunities for future work remain. First, it would be interesting to better integrate and improve ideas such as in Scaman and Virmaux (2019) to find improved bounds on gradients of Lipschitz-constrained neural networks, as this may allow to further reduce the amount of noise needed.

Second, various optimizations of the computational efficiency are possible. Currently one of the most important computational tasks is the computation of the spectral norm. Other approaches to more efficiently

compute or upper bound it can be explored. One obvious one is to use directly the $\ell_{2,1}$ and $\ell_{2,\infty}$ norms with our method, which would require to implement the exponential mechanism on the scaled gradient.

Our current work is limited to the application of our proposed method on feed-forward models for classification tasks and regression tasks with Lipschitz loss function. Although our method can be easily applied to some other tasks, the field remains open to extend it to other classes of models.

Finally, while our experiments have shown promising results, further theoretical analysis of Lip-DP-SGD, especially the interaction between sensitivity, learning rate and number of iterations, remains an interesting area of research, similar to the work of Song et al. (2020) on DP-SGD. An analysis on the interactions between hyperparameters would provide valuable insights into the optimal use of our method and its potential combination with other regularization techniques.

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

## A  Gradient clipping based DP-SGD

For comparison with Algorithm 1, Algorithm 2 shows the classic DP-SGD algorithm based on gradient clipping.

---

**Algorithm 2** DP-SGD: Differentially Private Stochastic Gradient Descent with gradient clipping.

---

**Input:** Data set $Z \in \mathcal{Z}^*$, model $f_\theta$, loss function $\mathcal{L}$, hypothesis space $\Theta \subseteq \mathbb{R}^k$, privacy parameters $\epsilon$ and $\delta$, noise multiplier $\sigma$, batch size $s \geq 1$, learning rate $\eta$, max gradient norm $C$
Initialize $\tilde{\theta}$ randomly from $\Theta$
**for** $t \in [T]$ **do**
    $V \leftarrow \emptyset$                                                                           ▷ Poisson sampling
    **while** $S = \emptyset$ **do**
        **for** $z \in Z$ **do**
            With probability $s/|Z|$: $V \leftarrow V \cup \{z\}$
        **end for**
    **end while**
    Draw $b_k \sim \mathcal{N}(0, \sigma^2 C^2 \mathbb{I})$
    **for** $i = 1 \dots |V|$ **do**                                                          ▷ Gradient clipping per sample
        $\tilde{g}_{k,i} \leftarrow \nabla_{\tilde{\theta}_k} \ell(f_{\tilde{\theta}}(x_i)) \min(1, C/\|\nabla_{\tilde{\theta}_k} \mathcal{L}(f(x_i; \tilde{\theta}))\|)$
    **end for**
    $\tilde{g}_k \leftarrow \frac{1}{|V|} \left( \sum_{i=1}^{|V|} \tilde{g}_{k,i} + b_k \right)$
    $\tilde{\theta}_k \leftarrow \tilde{\theta}_k - \eta(t) \tilde{g}_k$
**end for**
**Output:** $\tilde{\theta}$ and compute privacy cost $(\epsilon, \delta)$ with privacy accountant.

---

## B  Estimating Lipschitz values

We summarize the upper bounds of the Lipschitz values, either on the input or on the parameters, for each layer type in Table 2. It's important to mention that for the loss, the Lipschitz value is solely dependent on the output $x_{K+1}$.

| Layer | Definition | Lip. value on input $x_k$ | Lip. value on parameter $\theta_k$ |
|---|---|---|---|
| Dense | $\theta_k^\top x_k$ | $\|\theta_k\|$ | $\|x_k\|$ |
| Convolutional | $\theta_k * x_k$ | $\sqrt{h'w'}\|\theta_k\|$ | $\sqrt{h'w'}\|x_k\|$ |
| Normalization | $\frac{x_k^{(k:q)} - \mu^{(k:q)}}{\max(\alpha, \sigma^{(k:q)})}$ | $1/\alpha$ | - |
| ReLU | $\max(x_k, 0)$ | $1$ | - |
| Sigmoid | $\frac{1}{1+e^{-x_k}}$ | $1/2$ | - |
| Softmax Cross-entropy | $y \log \text{SOFTMAX}(x_{K+1})/\tau$ | $\sqrt{2}/\tau$ | - |
| Cosine Similarity | $\frac{x_{K+1}^\top y}{\|x_{K+1}\|_2 \|y\|_2}$ | $1/\min\|x_{K+1}\|$ | - |
| Multiclass Hinge | $\left\{ \max\left(0, \frac{m}{2} - x_{i_{K+1}} \cdot y_i\right) \right\}$ | $1$ | - |

Table 2: Summary table of Lipschitz values with respect to the layer.

with $\text{SOFTMAX}(x_i) = \frac{\exp(x_i)}{\sum_{j=1}^c \exp(x_j)}$, $c$ the number of classes. For cross-entropy, $\tau$ an hyperparameter on the Softmax Cross-entropy loss also known as the temperature. For convolutional layers, $h'$ and $w'$ are the height and width of the filter. For multiclass hinge, $m$ is a hyperparameter known as 'margin'.

### B.1 Details for the convolutional layer

**Theorem B.1.** *The convolved feature map $(\theta * \cdot) : \mathbb{R}^{n_k \times |x_k|} \to \mathbb{R}^{n_{k+1} \times n \times n}$, with zero or circular padding, is Lipschitz and*

$$\|\nabla_{\theta_k}(\theta_k * x_k)\|_p \leq \sqrt{h'w'}\|x_k\|_p \ \ and \ \ \|\nabla_{x_k}(\theta_k * x_k)\|_p \leq \sqrt{h'w'}\|\theta_k\|_p \tag{15}$$

*with $w'$ and $h'$ the width and the height of the filter.*

*Proof.* The output $x_{k+1} \in \mathbb{R}^{c_{out} \times n \times n}$ of the convolution operation is given by:

$$x_{k+1,c,r,s} = \sum_{d=0}^{c_{in}-1} \sum_{i=0}^{h'-1} \sum_{j=0}^{w'-1} x_{k,d,r+i,s+j} \theta_{k,c,d,i,j}$$

There follows, for any $p \in \{1, 2, +\infty\}$:

$$
\begin{aligned}
\|x_{k+1}\|_p^p &= \sum_{c=0}^{c_{out}-1} \sum_{r=1}^{n} \sum_{s=1}^{n} \left| \sum_{d=0}^{c_{in}-1} \sum_{i=0}^{h'-1} \sum_{j=0}^{w'-1} x_{k,d,r+i,s+j} \theta_{k,c,d,i,j} \right|^p \\
&\leq \sum_{c=0}^{c_{out}-1} \sum_{r=1}^{n} \sum_{s=1}^{n} \left( \sum_{d=0}^{c_{in}-1} \sum_{i=0}^{h'-1} \sum_{j=0}^{w'-1} |x_{k,d,r+i,s+j}|^p \right) \left( \sum_{d=0}^{c_{in}-1} \sum_{i=0}^{h'-1} \sum_{j=0}^{w'-1} |\theta_{k,c,d,i,j}|^p \right) \text{ triangle inequality, } p \in \{1, 2, +\infty\} \\
&= \left( \sum_{d=0}^{c_{in}-1} \sum_{i=0}^{h'-1} \sum_{j=0}^{w'-1} \sum_{r=1}^{n} \sum_{s=1}^{n} |x_{k,d,r+i,s+j}|^p \right) \left( \sum_{c=0}^{c_{out}-1} \sum_{d=0}^{c_{in}-1} \sum_{i=0}^{h'-1} \sum_{j=0}^{w'-1} |\theta_{k,c,d,i,j}|^p \right) \\
&\leq h'w' \left( \sum_{d=0}^{c_{in}-1} \sum_{r=1}^{n} \sum_{s=1}^{n} |x_{k,d,r,s}|^p \right) \left( \sum_{c=0}^{c_{out}-1} \sum_{d=0}^{c_{in}-1} \sum_{i=0}^{h'-1} \sum_{j=0}^{w'-1} |\theta_{k,c,d,i,j}|^p \right) \\
&= h'w'\|x_k\|_p^p \|\theta_k\|_p^p
\end{aligned}
$$

Since $\theta_k * \cdot$ is a linear operator:

$$\|(\theta_k * x_k) - (\theta'_k * x_k)\|_p = \|(\theta_k - \theta'_k) * x_k\|_p \leq \|\theta_k - \theta'_k\|_p (h'w')^{\frac{1}{p}} \|x_k\|_p$$

Finally, the convolved feature map is differentiable so the spectral norm of its Jacobian is bounded by its Lipschitz value:

$$\|\nabla_{\theta_k}(\theta_k * x_k)\|_p \leq (h'w')^{\frac{1}{p}} \|x_k\|_p$$

Analogously,

$$\|\nabla_{x_k}(\theta_k * x_k)\|_p \leq (h'w')^{\frac{1}{p}} \|\theta_k\|_p$$

$\square$

## C  Experiments

**Optimization.** For both tabular and image datasets, we employed Bayesian optimization Balandat et al. (2020). Configured as a multi-objective optimization program Daulton et al. (2020), our focus was to cover the Pareto front between model utility (accuracy) and privacy ($\epsilon$ values at a constant level of $\delta$, set to $1/n$ as has become common in this type of experiments). To get to finally reported values, we select the point on the pareto front given by the Python library BoTorch Balandat et al. (2020).

### C.1  Hyperparameters

**Hyperparameter selection.**   In the literature, there are a wide range of improvements possible over a direct application of SGD to supervised learning, including general strategies such as pre-training, data augmentation and feature engineering, and DP-SGD specific optimizations such as adaptive maximum gradient norm thresholds. All of these can be applied in a similar way to both Lip-DP-SGD and DP-SGD and to keep our comparison sufficiently simple, fair and understandable we didn't consider the optimization of these choices.

We did tune hyperparameters inherent to specific model categories, in particular the initial learning rate $\eta(0)$ (to start the adaptive learning rate strategy $\eta(t)$) and (for image datasets) the number of groups, and hyperparameters related to the learning algorithm, in particular the (expected) batch size $s$, the Lipschitz upper bound of the normalization layer $\alpha$ and the threshold $C$ on the gradient norm respectively weight norm.

The initial learning rate $\eta(0)$ is tuned while the following $\eta(t)$ are set adaptively. Specifically, we use the strategy of the Adam algorithm Kingma and Ba (2014), which update each parameter using the ratio between the moving average of the gradient (first moment) and the square root of the moving average of its squared value (second moment), ensuring fast convergence.

We also investigated varying the maximum norm of input vectors $X_0$ and the hyperparameter $\tau$ of the cross entropy objective function, but the effect of these hyperparameters turned out to be insignificant.

Both the clipping threshold $C$ for gradients in DP-SGD and the clipping threshold $C$ for weights in Lip-DP-SGD can be tuned for each layer separately. While this offers improved performance, it does come with the cost of consuming more of the privacy budget, and substantially increasing the dimensionality of the hyperparameter search space. In a few experiments we didn't see significant improvements in allowing per-layer varying of $C_k$, so we didn't further pursue this avenue.

Table 3 summarizes the search space of hyperparameters. It's important to note that we did not account for potential (small) privacy losses caused by hyperparameter search, a limitation also acknowledged in other recent works such as Papernot and Steinke (2022).

| Hyperparameter | Range |
|---|---|
| Noise multiplier $\sigma$ | [0.4, 5] |
| Weight clipping threshold $C$ | [1, 15] |
| Gradient clipping threshold $C$ | [1, 15] |
| Batch size $s$ | [32, 512] |
| $\eta(0)$ | [0.0001, 0.01] |
| Number of groups (group normalization) | [8, 32] |
| $\alpha$ (group normalization) | $[0.1/(|x_k^{(k:q)}|), 1/(|x_k^{(k:q)}|)]$ |

Table 3: Summary of hyperparameter space.

### C.2  Models

Table 4 shows details of the models we used to train on tabular and image datasets. We consider 7 tabular datasets: adult income Becker et al. (1996), android permissions Mathur et al. (2022), breast cancer Wolberg et al. (1995), default credit Yeh (2016), dropout Realinho et al. (2021), German credit Hofmann (1994) and nursery Rajkovic (1997). See Table 1 for the number of instances and features for each tabular dataset.

### C.3  Runtime

Our experiments didn't show significant deviations from the normal runtime behavior one can expect for neural network training. As an illustration, we compared on the MNIST dataset and on the CIFAR-10 dataset the median epoch runtime of DP-SGD with Lip-DP-SGD. Both implementations utilize Opacus

| Dataset | Image size | Model | Number of layers | Loss | No. of Parameters |
|---|---|---|---|---|---|
| Tabular Datasets | - | MLP | 2 | CE | 140 to 2,120 |
| MNIST | 28x28x1 | ConvNet | 4 | CE | 1,625,866 |
| FashionMNIST | 28x28x1 | ConvNet | 4 | CE | 1,625,866 |
| CIFAR-10 | 32x32x3 | WideResnet | 18 | CE | 11,176,842 |

Table 4: Summary table of datasets with respective models architectures details.

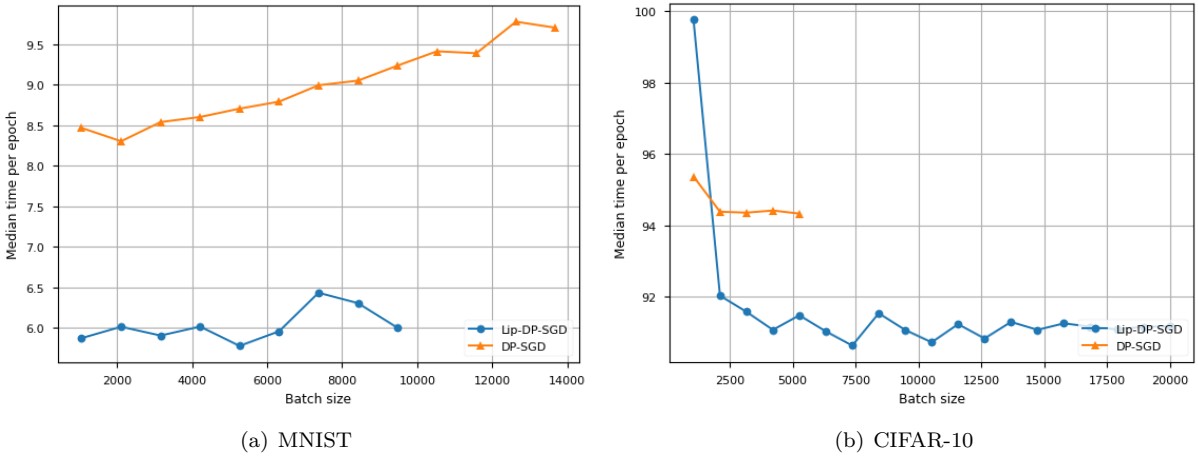

(a) MNIST  (b) CIFAR-10

Figure 2: Median runtime in seconds per batch size on one epoch over the MNIST dataset **??** and the CIFAR-10 dataset 2(b) comparing DP-SGD (in orange) and Lip-DP-SGD (in blue).

(Yousefpour et al., 2021) and PyTorch (Paszke et al., 2019), employing the same set of hyperparameters, such as augmentation multiplicity, to ensure a fair comparison. We measure runtime against the logical batch size, limiting the physical batch size to prevent memory errors as recommended by the PyTorch documentation (Paszke et al., 2019). Figure 2 shows how Lip-DP-SGD is more efficient in terms of runtime compared to DP-SGD, especially for big batch sizes. This is mainly due to the fact that DP-SGD requires to clip the gradient at the sample level, slowering down the process.

## C.4   Detailed results

Table 5 provides a summary of accuracy performances for tabular datasets with and without group normalization. It's worth noting that while epsilon values may not be identical across algorithms, we present the epsilon value of DP-SGD and report performances corresponding to lower epsilon values for the other two algorithms, consistent with Table 1.

## C.5   Gradient clipping behavior

In Section 3.4 we argued that DP-SGD introduces bias. There are several ways to demonstrate this. For illustration we show here the error between the true average gradient

$$g_k^{Lip-DP-SGD} = \frac{1}{|V|} \sum_{i=1}^{|V|} \nabla_{\tilde{\theta}_k} \ell(f_{\tilde{\theta}}(x_i))$$

i.e., the model update of Algorithm 1 without noise, and the average clipped gradient

$$g_k^{DP-SGD} = \frac{1}{|V|} \sum_{i=1}^{|V|} \text{clip}_C \left( \nabla_{\tilde{\theta}_k} \ell(f_{\tilde{\theta}}(x_i)) \right),$$

Table 5: Accuracy per dataset and method at $\epsilon = 1$ and $\delta = 1/n$, in bold the best result and underlined when the difference with the best result is not statistically significant at a level of confidence of 5%.

| Methods | | DP-SGD | | Lip-DP-SGD | | Fix-Lip-DP-SGD | |
|---|---|---|---|---|---|---|---|
| Datasets (#instances $n \times$ #features $p$) | $\epsilon$ | w/ GN | w/o GN | w/ GN | w/o GN | w/ GN | w/o GN |
| Adult Income (48842x14) Becker et al. (1996) | 0.414 | 0.822 | 0.824 | 0.829 | **0.831** | 0.713 | 0.804 |
| Android (29333x87) Mathur et al. (2022) | 1.273 | 0.947 | 0.951 | 0.952 | **0.959** | 0.701 | 0.945 |
| Breast Cancer (569x32) Wolberg et al. (1995) | 1.672 | 0.813 | 0.773 | **0.924** | 0.798 | 0.519 | 0.7 |
| Default Credit (30000x24) Yeh (2016) | 1.442 | 0.804 | 0.809 | 0.815 | **0.816** | 0.774 | 0.792 |
| Dropout (4424x36) Realinho et al. (2021) | 1.326 | 0.755 | 0.763 | 0.816 | **0.819** | 0.573 | 0.736 |
| German Credit (1000x20) Hofmann (1994) | 3.852 | 0.735 | 0.735 | 0.722 | 0.746 | 0.493 | 0.68 |
| Nursery (12960x8) Rajkovic (1997) | 1.432 | 0.916 | 0.919 | 0.912 | **0.931** | 0.487 | 0.89 |

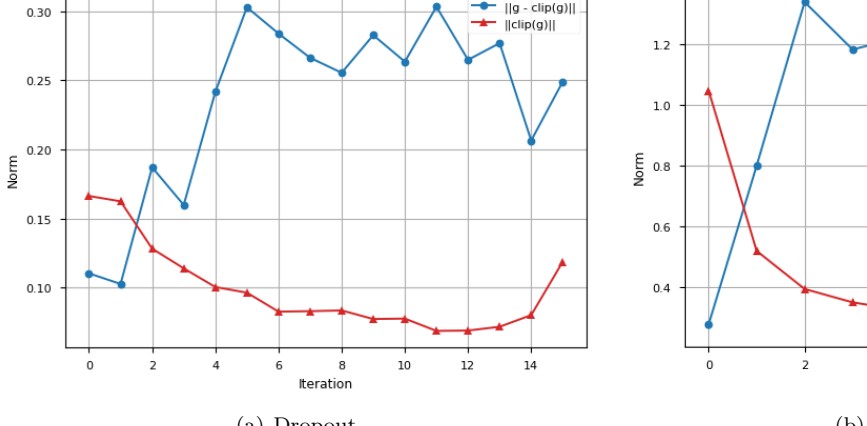

(a) Dropout                    (b) Adult Income

Figure 3: Norm of the average error $g - \text{clip}(g)$ (in blue) and norm of the average of $\text{clip}(g)$ (in red) across training iterations on the Dropout dataset 3(a) (averaged over 500 instances) and the Adult Income dataset 3(b) (averaged over 500 instances).

i.e., the model update of Algorithm 2 without noise.

Figure 3 shows the error $\left\| g_k^{Lip-DP-SGD} - g_k^{DP-SGD} \right\|$ together with the norm of the DP-SGD model update $\left\| g_k^{DP-SGD} \right\|$.

One can observe for both considered datasets that while the model converges and the average clipped gradient decreases, the error between DP-SGD's average clipped gradient and the true average gradient increases. At the end, the error in the gradient caused by clipping is significant, and hence the model converges to a different point than the real optimum.

## D   Lip-DP-SGD library

We offer an open-source toolkit for implementing LipDP-SGD on any FNN model structure. This toolkit builds on the Opacus and PyTorch libraries. Drawing inspiration from Opacus, our library is based on two main components: the 'DataLoader', which utilizes Poisson sampling to harness the advantages of privacy amplification Kasiviswanathan et al. (2011), and the 'Optimizer', responsible for sensitivity calculation, differential privacy noise addition, and parameter normalization during each iteration.

'README.md', provided in the supplementary materials, details how to run the library and how to reproduce the experiments.

# E Avoiding the bias of gradient clipping

We show that Lip-DP-SGD converges to a local minimum in $\Theta_{\leq C}$ while DP-SGD suffers from bias and may converge to a point which is not a local minimum of $\Theta$.

We use the word 'converge' here somewhat informally, as in each iteration independent noise is added the objective function slightly varies between iterations and hence none of the mentioned algorithms converges to an exact point. We here informally mean approximate convergence to a small region, assuming a sufficiently large data set $Z$ and/or larger $\epsilon$ such that privacy noise doesn't significantly alter the shape of the objective function. Our argument below hence makes abstraction of the noise for simplicity, but in the presence of small amounts of noise a similar argument holds approximately, i.e., after sufficient iterations Lip-DP-SGD will produce $\theta$ values close to a locally optimal $\theta^*$ while DP-SGD may produce $\theta$ values in a region not containing the relevant local minimum.

First, let us consider convergence.

**Theorem** 3.2. Let $F$ be an objective function as defined in Section 2.2, and $Z$, $f_\theta$, $\mathcal{L}$, $\Theta = \Theta_{\leq C}$, $T$, $\sigma = 0$, $s$, $\eta$ and $C$ be input parameters of Lip-DP-SGD satisfying the requirements specified in Section 3.3. Assume that for these inputs Lip-DP-SGD converges to a point $\theta^*$ (in the sense that $\lim_{k,T \to \infty} \theta_k = \theta^*$). Then, $\theta^*$ is a local optimum of $F(\theta, Z)$ in $\Theta_{\leq C}$.

*Proof sketch.* We consider the problem of finding a local optimum in $\Theta_{\leq C}$:

$$\begin{aligned} \text{minimize} \quad & F(\theta, Z) \\ \text{subject to} \quad & \|\theta\|_2 \leq C \end{aligned}$$

We introduce a slack variable $\zeta$:

$$\begin{aligned} \text{minimize} \quad & F(\theta, Z) \\ \text{subject to} \quad & \|\theta\|_2 + \zeta^2 = C \end{aligned}$$

Using Lagrange multipliers, we should minimize

$$F(\theta, Z) - \lambda(\|\theta\|_2 + \zeta^2 - C)$$

An optimum in $\theta$, $\lambda$ and $\zeta$ satisfies

$$\begin{aligned} \nabla_\theta F(\theta, Z) - \lambda\theta &= 0 & (16) \\ \|\theta\|_2 + \zeta^2 - C &= 0 & (17) \\ 2\lambda\zeta &= 0 & (18) \end{aligned}$$

From Eq 18, either $\lambda = 0$ or $\zeta = 0$ If $\zeta > 0$, $\theta$ is in the interior of $\Theta_{\leq C}$ and there follows $\lambda = 0$ and from Eq 16 that $\nabla_\theta F(\theta, Z) = 0$. For such $\theta$, Lip-DP-SGD does not perform weight clipping. If the learning rate is sufficiently small, and if it converges to a $\theta$ with norm $\|\theta\|_2 < C$ it is a local optimum. On the other hand, if $\zeta = 0$, there follows from Eq 17 that $\|\theta\|_2 = C$, i.e., $\theta$ is on the boundary of $\Theta_{\leq C}$. If $\theta$ is a local optimum in $\Theta_{\leq C}$, then $\nabla_\theta F(\theta, Z)$ is perpendicular on the ball of vectors $\theta$ with norm $C$, and for such $\theta$ Lip-DP-SGD will add the multiple $\eta(t).\nabla_\theta F(\theta, Z)$ to $\theta$ and will next scale $\theta$ back to norm $C$, leaving $\theta$ unchanged. For a $\theta$ which is not a local optimum in $\Theta_{\leq C}$, $\nabla_\theta F(\theta, Z)$ will not be perpendicular to the ball of $C$-norm parameter vectors, and adding the gradient and brining the norm back to $C$ will move $\theta$ closer to a local optimum on this boundary of $\Theta_{\leq C}$. This is consistent with Eq 16 which shows the gradient with respect to $\theta$ for the constrained problem to be of the form $\nabla_\theta F(\theta, Z) - \lambda\theta$. $\square$

Theorem 3.2 shows that in a noiseless setting, if Lip-DP-SGD converges to a stable point that point will be a local optimum in $\Theta_{\leq C}$. In the presence of noise and/or stochastic batch selection, algorithms of course don't converge to a specific point but move around close to the optimal point due to the noise in each iteration, and advanced methods exist to examine such kind of convergence. The conclusion remains the same: Lip-DP-SGDwill converge to a neighborhood of the real local optimum, while as we argue DP-SGD will often converge to a neighborhood of a different point.

Second, we argue that DP-SGD introduces bias. This was already pointed out in Chen et al. (2020)'s examples 1 and 2. In Section C.5 we also showed experiments demonstrating this phenomenon. Below, we provide a simple example which we can handle (almost) analytically.

A simple situation where bias occurs and DP-SGD does not converge to an optimum of $F$ is when errors aren't symmetrically distributed, e.g., positive errors are less frequent but larger than negative errors.

Consider the scenario of simple linear regression. A common assumption of linear regression is that instances are of the form $(x_i, y_i)$ where $x_i$ is drawn from some distribution $P_x$ and $y_i = ax_i + b + e_i$ where $e_i$ is drawn from some zero-mean distribution $P_e$. When no other evidence is available, one often assume $P_e$ to be Gaussian, but this is not necessarily the case. Suppose for our example that $P_x$ is the uniform distribution over $[0, 1]$ and $P_e$ only has two possible values, in particular $P_e(9) = 0.1$, $P_e(-1) = 0.9$ and $P_e(e) = 0$ for $e \notin \{9, -1\}$. So with high probability there is a small negative error $e_i$ while with small probability there is a large positive error, while the average $e_i$ is still 0. Consider a dataset $Z = \{(x_i, y_i)\}_{i=1}^n$. Let us consider a model $f(x) = \theta_1 x \theta_2$ and let us use the square loss $\mathcal{L}(\theta, Z) = \sum_{i=1}^n \ell(x_i, y_i)/n$ with $\ell(\theta, x, y) = (\theta_1 x + \theta_2 - y)^2$. Then, the gradient is

$$\nabla_\theta \ell(\theta, x, y) = (2(\theta_1 x + \theta_2 - y)x, 2(\theta_1 x + \theta_2 - y))$$

For an instance $(x_i, y_i)$ with $y_i = ax_i + b + e_i$, this implies

$$\nabla_\theta \ell(\theta, x_i, y_i) = (2((\theta_1 - a)x_i + (\theta_2 - b) - e_i)x_i, 2((\theta_1 - a)x_i + (\theta_2 - b) - e_i))$$

For sufficiently large datasets $Z$ where empirical loss approximates population loss, the gradient considered by Lip-DP-SGD will approximate

$$
\begin{aligned}
\nabla_\theta \mathcal{L}(\theta, Z) &\approx \sum_{e \in \{10,\}} P_e(e) \int_0^1 \nabla_\theta \ell(\theta, x, ax + b + e) \mathrm{d}x \\
&= \sum_{e \in \{10,\}} P_e(e) \int_0^1 (2((\theta_1 - a)x + (\theta_2 - b) - e)x, 2((\theta_1 - a)x + (\theta_2 - b) - e)) \, \mathrm{d}x \\
&= \int_0^1 (2((\theta_1 - a)x^2 + (\theta_2 - b)x - x\mathbb{E}[e]), 2((\theta_1 - a)x + (\theta_2 - b) - \mathbb{E}[e])) \, \mathrm{d}x \\
&= (2((\theta_1 - a)/3 + (\theta_2 - b)/2), 2((\theta_1 - a)/2 + (\theta_2 - b)))
\end{aligned}
$$

This gradient becomes zero if $\theta_1 = a$ and $\theta_2 = b$ as intended.

However, if we use gradient clipping with threshold $C = 1$ as in DP-SGD, we get:

$$
\begin{aligned}
\tilde{g} &\approx \sum_{e \in \{10,\}} P_e(e) \int_0^1 clip_1 \left( \nabla_\theta \ell(\theta, x, ax + b + e) \right) \mathrm{d}x \\
&= \sum_{e \in \{10,\}} P_e(e) \int_0^1 clip_1 \left( (2((\theta_1 - a)x + (\theta_2 - b) - e)x, 2((\theta_1 - a)x + (\theta_2 - b) - e)) \right) \mathrm{d}x
\end{aligned}
$$

While for a given $e$ for part of the population $(\theta_1 - a)x + \theta_2 - b$ may be small, for a fraction of the instances the gradients are clipped. For the instances with $e = 9$ this effect is stronger. The result is that for $\theta_1 = a$ and $\theta_2 = b$ the average clipped gradient $\tilde{g}$ doesn't become zero anymore, in particular $\|\tilde{g}\| = 0.7791$. In fact, $\tilde{g}$ becomes zero for $\theta_1 = a + 0.01765$ and $\theta_2 = b + 0.94221$. Figure E illustrates this situation.

Figure 4: An example of gradient clipping causing bias, here the average gradient becomes zero at $(0, 0)$ while the average clipped gradient is 0 at another point, causing convergence of DP-SGD to that point rather than the correct one.

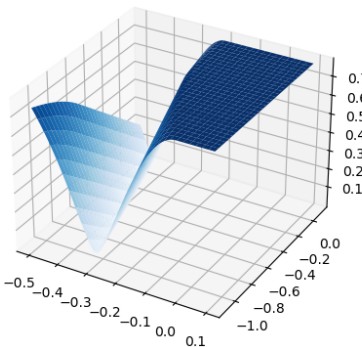

