# OpenReview forum: "DP-SGD with weight clipping"
_TMLR — Under review for TMLR_

### Review · Reviewer_mpgo · 2026-06-29

**Summary Of Contributions:**

This submission proposes Lip-DP-SGD, a variant of DP-SGD that aims to avoid per-sample gradient clipping by using weight clipping and Lipschitz-constrained neural networks. The authors derive layer-wise Lipschitz bounds and use them to estimate gradient sensitivity. They argue that this allows Gaussian noise calibration without traditional gradient clipping and reduces clipping bias.

**Audience:**

Yes

**Audience Explanation:**

The topic is relevant to a subset of the TMLR audience. Differentially private deep learning, gradient clipping bias, per-example clipping overhead, and Lipschitz-constrained neural networks are active research areas. The idea of replacing gradient clipping with parameter-space or Lipschitz constraints is potentially useful, and the empirical direction may motivate further work.

**Broader Impact Concerns:**

No.

**Claims And Evidence:**

No

**Claims Explanation:**

The key issue is that the proposed sensitivity calibration uses a batch-dependent quantity $X_k$, defined as the maximum norm of the layer input over the sampled minibatch. This quantity depends on private data. Moreover, it is used as a shared denominator for all examples in the batch. Therefore, when one record changes, the denominator can change and consequently rescale the contributions of all other records. Bounding each individually scaled gradient by a constant does not imply a valid bound on the sensitivity of the scaled sum or average.

There is also a mismatch between the theoretical bound in Eq. (14), which uses a product over layers $i=k+1,\ldots,K$, and Algorithm 1, which uses a product over $i=k,\ldots,K$. This can lead to zero or underestimated noise when the current layer weight norm is small, even though the parameter gradient may be nonzero.

The treatment of bias parameters is also incomplete. The derivation for dense layers requires an augmented input vector $(x,1)$, but Algorithm 1 uses only the norm of the layer input. This can fail to bound bias gradients and can even produce division by zero when the layer input norm is zero.

The privacy accounting is incomplete as well. The algorithm adds independent Gaussian noise per layer but then uses the same accounting as standard DP-SGD, without proving a valid joint sensitivity bound for the concatenated multilayer gradient. The Poisson sampling procedure is also modified by resampling until a non-empty minibatch is obtained, while the privacy accountant is described as if standard Poisson subsampling were used.

The convergence claim is also overstated. The proof sketch for Theorem 3.2 does not establish convergence to a local optimum; at most it suggests a projected first-order stationarity condition under restrictive assumptions. The Lagrangian derivation itself is inconsistent with the stated norm constraint, and the argument does not handle nonconvexity, stochastic gradients, or DP noise.

The empirical results are interesting but not sufficient to support the paper’s main claims. The authors perform Bayesian hyperparameter optimization but explicitly do not account for the privacy cost of hyperparameter tuning. This weakens all reported privacy-utility comparisons. Some experimental statements are also inconsistent with the tables, e.g., the claim that Fix-Lip-DP-SGD is inferior even to DP-SGD is contradicted by the German Credit row in Table 1.

**Requested Changes:**

The current paper has serious issues:

1. The central sensitivity quantity $X_k$ is batch-dependent and data-dependent.


2. The proof of differential privacy does not handle the sensitivity of the data-dependent scaled sum.


3. Algorithm 1 and Eq. (14) use inconsistent layer products for the noise scale.


4. Bias parameters are not correctly handled in the sensitivity bound.


5. The per-layer Gaussian mechanism and the modified Poisson sampling procedure are not correctly accounted for.


6. Hyperparameter tuning privacy cost is not included.


7. Several mathematical claims about Lipschitz constants and convergence are overstated or incorrect.

8. There are so many typos and errors even with ``??" marks. The presentations are substandard.

---

### Review · Reviewer_igRR · 2026-07-02

**Summary Of Contributions:**

This paper proposes Lip-DP-SGD, a differentially private training method that aims to avoid per-sample gradient clipping by using Lipschitz-constrained neural networks and layerwise sensitivity estimates. The algorithm clips or normalizes model weights, tracks layer input norms, scales gradients layer by layer, and adds Gaussian noise using an RDP accountant. The paper argues that this avoids the optimization bias induced by standard gradient clipping, proves a noiseless convergence result for the constrained problem, and reports improved empirical performance over DP-SGD baselines on tabular datasets and on MNIST, Fashion-MNIST, and CIFAR-10.
The main strengths are that the paper targets an important weakness of DP-SGD, namely clipping bias, and that it connects differential privacy with the literature on Lipschitz-constrained neural networks. The empirical results, if fully supported, would be interesting to researchers working on private deep learning. The main weaknesses are that the central privacy claim is not yet convincingly established, the experimental artifact does not fully match the paper's description. The paper should also better isolate what is new beyond closely related work on DP-SGD without clipping using Lipschitz neural networks.

**Audience:**

Yes

**Audience Explanation:**

Differentially private deep learning is clearly within TMLR's scope, and the bias introduced by gradient clipping is an important issue for DP-SGD. A method that can reduce or remove per-sample clipping while preserving valid privacy guarantees would be of interest to researchers studying private optimization, private image classification, and Lipschitz-constrained neural networks.
The paper's direction is therefore interesting even though the current evidence is not sufficient for acceptance. If the authors can repair the privacy proof, align the implementation with the paper, and provide clearer controlled comparisons, the findings could be useful to at least a subset of the TMLR audience.

**Broader Impact Concerns:**

The paper studies privacy-preserving learning and therefore has potentially positive broader impact. My main concern is that privacy guarantees may be overstated before the DP argument is fully established. If practitioners rely on a data-dependent sensitivity estimate that is not correctly accounted for, they may believe they have a formal privacy guarantee when they do not.

The authors should include or strengthen a broader impact statement that discusses this risk and clearly states the assumptions under which the privacy guarantee holds.

**Claims And Evidence:**

No

**Claims Explanation:**

My main concern is the privacy argument. Algorithm 1 computes a layerwise quantity $X_k$ as the maximum norm of layer inputs over the sampled batch, and the supplementary implementation similarly computes max input norms from the current batch via MaxInputNormLayer before scaling gradients and adding Gaussian noise. The paper states that the scaled gradient sensitivity depends only on privatized parameters and not on the data, but this does not follow from the algorithm as written.

More specifically, the paper bounds the norm of each sample's scaled gradient individually, but it does not analyze the sensitivity of the full aggregate gradient mechanism. For adjacent datasets or adjacent sampled batches, the max input norm $X_k$ can change, which means that even samples shared by both batches may be scaled by different data-dependent denominators. The DP proof needs to bound the full mechanism, including this data-dependent gradient transformation. The current proof of Theorem 3.1 does not address this issue, and this is a central gap because the paper's main contribution is a DP mechanism.

The convergence claim is also limited. Theorem 3.2 assumes $sigma = 0$, so it is a noiseless constrained-optimization statement rather than a convergence guarantee for the actual private noisy algorithm. This is useful context, but the paper should not let it appear to establish behavior of the DP algorithm without further argument.

**Requested Changes:**

Critical to securing my recommendation for acceptance:

1. Provide a complete DP proof for the full aggregate mechanism. The proof should bound $||g(Z) - g(Z')||$ for adjacent datasets or adjacent sampled batches, explicitly accounting for the fact that $X_k$ is computed from the private batch and can change between adjacent inputs. Bounding individual scaled sample gradients is not sufficient unless the authors also show that the data-dependent scaling of all samples in the aggregate does not increase sensitivity or is otherwise accounted for.

2. Clarify or revise the use of $X_k$. If the current batch-dependent $X_k$ is not covered by the DP proof, the authors should replace it with a public/global bound, privatize it with separate privacy accounting, or narrow the claim accordingly.

3. Clarify how the normalization layers used in experiments satisfy the paper's Lipschitz assumptions. In particular, explain the relation between the paper's modified group normalization and the standard Flax 'GroupNorm' used in the supplementary implementation, or update the implementation/paper so they match.

4. Provide a controlled comparison against closely related Lipschitz-DP-SGD work, especially DP-SGD without clipping via Lipschitz neural networks. The comparison should isolate the contribution of allowing $||\theta_k|| < C$, the current/layerwise input-norm scaling, and other orthogonal improvements such as augmentation multiplicity, architecture choices, and optimizer/tuning differences.

5. Clearly state that Theorem 3.2 is a noiseless result and discuss whether, and in what sense, it extends to the noisy private training setting.

Would strengthen the work:

1. Add ablations separating the effects of weight clipping or spectral normalization, max-input-norm scaling, augmentation multiplicity, and optimizer choice.

2. Include statistical uncertainty for image experiments in a numeric table, not only in plots.

3. Improve notation and writing. Several equations are difficult to parse, and there are avoidable typos such as 'an novel', 'constrains', 'paremeter', and 'librairy'.

4. State clearly whether the method uses add/remove adjacency or replace-one adjacency, and keep that convention consistent through the privacy accountant and experiments.

5. Clarify whether the reported code is intended as a reference implementation of the exact algorithm in the paper or only as a partial reproduction package.

---

### Review · Reviewer_qC69 · 2026-07-13

**Summary Of Contributions:**

This paper proposes **Lip-DP-SGD**, a differentially private training algorithm that aims to avoid the optimization bias introduced by per-example gradient clipping in standard DP-SGD. Instead of clipping gradients, the method enforces Lipschitz constraints through weight clipping, estimates layer-wise gradient sensitivity using bounds on the Lipschitz constants of the network, scales the layer-wise gradients by a maximum activation norm, and then adds Gaussian noise calibrated to the estimated sensitivity.

The main claimed contributions are:

1. a method for estimating gradient sensitivity in feed-forward neural networks using Lipschitz-constrained layers;
2. a new algorithm, Lip-DP-SGD, that replaces gradient clipping with weight clipping and layer-wise sensitivity estimation;
3. an argument that this avoids the bias introduced by standard gradient clipping;
4. empirical evaluations on MNIST, Fashion-MNIST, CIFAR-10, and several tabular datasets, where the proposed method is reported to outperform DP-SGD and a fixed-norm variant.

The motivation is strong and the topic is important. Gradient clipping is a known source of bias in DP-SGD, and a correct method that avoids per-example clipping while preserving rigorous differential privacy guarantees would be valuable. The empirical results are also promising.

However, I have serious concerns about the correctness and completeness of the privacy analysis. In particular, the algorithm uses a data-dependent scaling quantity $X_k$, but the proof does not appear to account for the sensitivity introduced by this data-dependent denominator. The paper also does not clearly justify privacy composition across layers, and there is an apparent inconsistency between the sensitivity bound in Equation (14) and the definition of $\Delta_k$ in Algorithm 1. Because the central claim of the paper is a differentially private algorithm, these issues are critical.

**Additional Comments:**

I appreciate the authors' attempt to address the bias introduced by gradient clipping in DP-SGD. The direction is interesting, and the empirical results suggest that Lipschitz-based sensitivity estimation may be a promising approach.

My main concern is not the motivation or the empirical ambition of the paper, but the soundness of the privacy analysis. Since the primary contribution is a differentially private algorithm, the proof needs to be airtight. In particular, the data-dependent scaling term $X_k$ should be treated as a central part of the privacy analysis rather than as a benign normalization factor.

If the authors can provide a valid sensitivity proof for the full query, clarify the layer-wise privacy accounting, and make the experimental privacy accounting complete, the paper could become a strong contribution.

**Audience:**

Yes

**Audience Explanation:**

Yes. The paper addresses an important problem in differentially private machine learning: the utility loss and optimization bias caused by gradient clipping in DP-SGD. Many researchers in TMLR's audience work on privacy-preserving machine learning, optimization, generalization, and robust or Lipschitz-constrained neural networks. A correct and practical alternative to per-example gradient clipping would be of clear interest to this audience.

The proposed connection between Lipschitz-constrained networks and sensitivity estimation for private optimization is also interesting. The empirical results, especially on CIFAR-10 and on the tabular datasets, suggest that the approach may have practical value if the privacy argument can be made rigorous.

Thus, even though I do not think the paper is ready for acceptance in its current form, the direction is relevant and potentially impactful.

**Broader Impact Concerns:**

The paper is about differential privacy, so its intended broader impact is positive: improving the utility of privacy-preserving machine learning could make it easier to train models on sensitive data while protecting individuals.

However, the main broader impact concern is that an incomplete or incorrect privacy proof could give practitioners a false sense of privacy protection. If the data-dependent scaling factor $X_k$, layer-wise composition, hyperparameter tuning, or spectral norm approximation are not properly accounted for, then the deployed method may not satisfy the claimed $(\epsilon,\delta)$-DP guarantee.

I recommend that the authors include a broader impact discussion emphasizing the assumptions required for the privacy guarantee, the risks of using the method outside those assumptions, and the privacy cost of model selection and hyperparameter tuning. This is especially important because the method is evaluated on tabular datasets that resemble realistic sensitive-data applications.

**Claims And Evidence:**

No

**Claims Explanation:**

I do not think the main claims are currently supported by sufficiently accurate, convincing, and clear evidence.

The empirical evidence is promising, but the central theoretical claim is that Lip-DP-SGD satisfies differential privacy. I do not find the current privacy proof convincing. In Algorithm 1, the method computes

$$
X_k = \max_i \| f_{\tilde{\theta}, k-1}(x_i) \|_2
$$

over the sampled private mini-batch and then uses $1/X_k$ to scale the layer-wise gradient before adding Gaussian noise. This makes the query data-dependent not only through the gradients but also through the denominator. For neighboring batches or neighboring datasets, $X_k$ may change, and this change rescales the contributions of all examples in the batch, not only the differing example.

The paper gives a pointwise bound of the form

$$
\left\|
\frac{\nabla_{\tilde{\theta}_k} \ell(f_{\tilde{\theta}}(x),y)}{X_k}
\right\|_{2,p}
\leq
\tau \prod_{i=k+1}^K c_i,
$$

but this does not by itself establish a valid global sensitivity bound for the full query

$$
V \mapsto
\frac{1}{X_k(V)}
\sum_{i \in V}
\nabla_{\theta_k} \ell(f_\theta(x_i), y_i).
$$

The data dependence of $X_k$ must be accounted for explicitly. Without this, the privacy proof is incomplete.

A second issue is that the algorithm adds independent Gaussian noise to each layer using layer-wise sensitivities \(\Delta_k\), while all layer gradients are computed from the same private mini-batch. The privacy analysis should either bound the sensitivity of the full concatenated update

$$
\tilde{g} = (\tilde{g}_1, \ldots, \tilde{g}_K)
$$

or provide an explicit composition argument across layers. This is not clearly done.

There is also an apparent inconsistency between Equation (14), which suggests a bound involving

$$
\tau \prod_{i=k+1}^K c_i,
$$

and Algorithm 1, which defines

$$
\Delta_k = \tau \prod_{i=k}^K c_i.
$$

This indexing difference matters because $c_k$ can be smaller than $1$, so including it could under-estimate the required noise.

Finally, some supporting derivations, including the convolutional and normalization-layer Lipschitz bounds, require more rigor. The experimental section also does not account for privacy loss from hyperparameter tuning, even though Bayesian optimization is used over important hyperparameters. Therefore, while the empirical results are interesting, the current evidence does not convincingly support the paper's strongest claims.

**Requested Changes:**

I list the requested changes below. I mark each item as either **critical for acceptance** or **would strengthen the work**.

### 1. Provide a complete privacy proof for the data-dependent scaling factor $X_k$
**Critical for acceptance.**

The most important issue is the data dependence of
$
X_k = \max_i \| f_{\tilde{\theta}, k-1}(x_i) \|_2.
$

Since $X_k$ is computed from the sampled private mini-batch, the query being privatized is not simply a sum of bounded per-example gradients. It is a ratio in which the denominator can change between neighboring datasets. The authors should provide a formal sensitivity proof for the full query
$
V \mapsto
\frac{1}{X_k(V)}
\sum_{i \in V}
\nabla_{\theta_k} \ell(f_\theta(x_i), y_i).
$

The proof should account for the fact that changing one example can change $X_k$, which can rescale all terms in the batch. If such a proof is not possible, the algorithm should be modified to use a public bound, a privately released bound with privacy cost included, or another normalization scheme whose sensitivity can be rigorously bounded.

### 2. Provide a joint privacy analysis across layers
**Critical for acceptance.**

Algorithm 1 releases or uses noisy gradients for all layers, and all layer-wise gradients are computed from the same private mini-batch. The privacy proof should therefore analyze the full released update vector

$$
\tilde{g} = (\tilde{g}_1, \ldots, \tilde{g}_K),
$$

or provide a valid composition argument over the layer-wise mechanisms.

If the authors rely on per-layer Gaussian mechanisms, they should state exactly how privacy loss composes across layers within each iteration. Alternatively, they should calibrate noise to the sensitivity of the concatenated gradient vector, for example involving a quantity such as

$$
\left( \sum_{k=1}^K \Delta_k^2 \right)^{1/2},
$$

unless a tighter joint bound is proved.

### 3. Resolve the discrepancy between Algorithm 1 and Equation (14)
**Critical for acceptance.**

Equation (14) suggests that the sensitivity for layer $k$ should involve

$$
\tau \prod_{i=k+1}^K c_i,
$$

whereas Algorithm 1 defines

$$
\Delta_k = \tau \prod_{i=k}^K c_i.
$$

This difference is not merely cosmetic. Since $c_k = \min(C, \|\theta_k\|)$, it may be smaller than $1$. Including $c_k$ could under-estimate the required noise relative to the stated bound. The authors should clarify whether Algorithm 1, Equation (14), or the definition of $c_k$ is incorrect, and revise the privacy proof accordingly.

### 4. Make the Lipschitz and norm derivations rigorous for the exact norms and layers used
**Critical for acceptance.**

The privacy guarantee depends directly on the claimed Lipschitz and sensitivity bounds. These derivations should be made fully rigorous for the exact implementation used in the experiments, especially the $\ell_{2,2}$ norm.

In particular:

- the convolutional-layer bound should be stated and proved for the exact norm used;
- the treatment of $p=\infty$ should avoid expressions involving $p$-powers that are not meaningful in that case;
- the normalization-layer bound should explicitly account for the derivative of the mean and variance terms;
- any constants such as $\sqrt{h'w'}$ should be justified carefully;
- the proof should distinguish clearly between bounds with respect to inputs and bounds with respect to parameters.

### 5. Account for approximation error in spectral norm estimation
**Critical for acceptance.**

The paper states that the $\ell_{2,2}$ norm is estimated using power-method-based techniques. If the privacy proof requires an upper bound on the spectral norm, then numerical approximation error must be handled conservatively. The authors should explain how they ensure that the computed norm is an upper bound rather than an underestimate. If the estimate can be below the true spectral norm, the privacy guarantee may not hold.

### 6. Account for privacy loss from hyperparameter tuning or change the experimental protocol
**Critical for acceptance.**

The experiments use Bayesian optimization over important hyperparameters, including the noise multiplier, batch size, learning rate, clipping or weight thresholds, and normalization parameters. The paper states that potential privacy losses from hyperparameter search are not accounted for.

For a DP paper, this is a significant limitation. The authors should either:

- use only public validation data for hyperparameter tuning;
- account for private validation and model selection using a valid DP method;
- report the additional privacy cost of hyperparameter optimization;
- or clearly mark the tuned results as not satisfying the final claimed $(\epsilon,\delta)$-DP budget.

### 7. Provide complete and matched experimental reporting
**Critical for acceptance.**

The experimental section should provide complete tables for all datasets and methods. These tables should include:

- exact $\epsilon$ and $\delta$;
- noise multiplier;
- batch size;
- number of epochs;
- learning rate;
- clipping threshold or weight threshold;
- model architecture;
- number of random seeds;
- number of hyperparameter tuning trials;
- mean accuracy;
- standard deviation or standard error;
- whether group normalization and other regularization techniques were used.

The comparison should make clear whether all methods are matched in privacy budget, architecture, tuning budget, training epochs, and validation protocol.

### 8. Weaken or rigorously prove the convergence/local-optimum claim
**Critical for acceptance.**

Theorem 3.2 claims that, without DP noise, Lip-DP-SGD converges to a local optimum of the constrained objective if the iterates converge. This seems stronger than what is currently justified.

The algorithm uses stochastic mini-batches, layer-wise scaling by $1/X_k$, projection/weight clipping, and in experiments an Adam-style adaptive update. Even if the iterates converge, more assumptions are needed to conclude convergence to a local optimum rather than to a stationary point or to a fixed point of a modified projected stochastic method.

The authors should either provide a rigorous proof under precise assumptions or weaken the statement.

### 9. Add sensitivity sanity checks or controlled experiments
**Would strengthen the work.**

It would be useful to include controlled experiments verifying that the claimed sensitivity bounds are conservative in practice. For example, the authors could numerically compare the claimed upper bounds with observed differences between gradients on neighboring mini-batches. This would not replace a formal proof, but it would improve confidence in the method.

### 10. Clarify the comparison with Fix-Lip-DP-SGD and prior Lipschitz-DP methods
**Would strengthen the work.**

The comparison with the fixed-norm variant is useful. The paper would be stronger if it more clearly explained when and why allowing $\|\theta_k\| < C$ reduces noise and improves utility. It would also help to clarify the precise difference between this method and prior work on DP-SGD without clipping using Lipschitz neural networks.

### 11. Improve notation, algorithm presentation, and writing
**Would strengthen the work.**

Several notation and presentation issues should be corrected. For example:

- the DP definition should use $P(A(Z_1) \in S)$, not $P(A(Z_1) \subseteq S)$;
- Algorithm 2 appears to use both $S$ and $V$ for the sampled batch;
- the notation $c_k$, $X_k$, $\Delta_k$, and $V_k$ should be made consistent;
- the distinction between model parameters, privatized parameters, and post-processed parameters should be clarified.

The paper also contains several typos and grammar issues, such as “an novel,” “developped,” “dependance,” “pre-definied,” and “slowering.”